# The Immunomodulatory Role of Regulatory T Cells in Preterm Birth and Associated Pregnancy Outcomes

**DOI:** 10.3390/ijms252211878

**Published:** 2024-11-05

**Authors:** Nicoleta Mureanu, Amanda M. Bowman, Imogen A. Porter-Wright, Priya Verma, Athina Efthymiou, Kypros H. Nicolaides, Cristiano Scotta, Giovanna Lombardi, Rachel M. Tribe, Panicos Shangaris

**Affiliations:** 1School of Life Course & Population Sciences, King’s College London, 10th Floor North Wing, St Thomas’ Hospital, London SE1 7EH, UK; n.mureanu@nhs.net (N.M.); amanda.bowman@kcl.ac.uk (A.M.B.); a.efthymiou@nhs.net (A.E.); kypros@fetalmedicine.com (K.H.N.); rachel.tribe@kcl.ac.uk (R.M.T.); 2Harris Birthright Research Centre for Fetal Medicine, King’s College London, London SE1 7EH, UK; 3Faculty of Medicine, Department of Obstetrics and Gynaecology, Carol Davila University of Medicine and Pharmacy, Bulevardul Eroii Sanitari 8, 050474 Bucharest, Romania; 4Peter Gorer Department of Immunobiology, School of Immunology & Microbial Sciences, Faculty of Life Sciences & Medicine, King’s College London, London SE1 7EH, UK; i.porter-wright@smd23.qmul.ac.uk (I.A.P.-W.); priya.g.verma@kcl.ac.uk (P.V.); cristiano.scotta@brunel.ac.uk (C.S.); giovanna.lombardi@kcl.ac.uk (G.L.); 5Department of Life Sciences, Centre for Inflammation Research and Translational Medicine, Brunel University London, London UB8 3PH, UK

**Keywords:** spontaneous preterm birth, Tregs, pregnancy immunology

## Abstract

Spontaneous preterm birth (sPTB), defined as live birth before 37 weeks of gestational age, is associated with immune dysregulation and pro-inflammatory conditions that profoundly impact newborn health. The question of immune integrity at the maternal-foetal interface is a focus of recent studies centring not only sPTB but the conditions often affiliated with this outcome. Regulatory T cells (Tregs) play a critical anti-inflammatory role in pregnancy, promoting foetal tolerance and placentation. Due to this gestational role, it is hypothesised that decreased or dysfunctional Tregs may be implicated in cases of sPTB. This review examines studies comparing Treg presence in healthy term pregnancies and those with sPTB-associated conditions. Conflicting findings across different conditions and within sPTB itself have been identified. However, notable findings from the research indicate increased proinflammatory cytokines in pregnancies suffering from premature rupture of membranes (pPROM), chorioamnionitis, infection, preeclampsia, and gestational diabetes (GDM). Additionally, reduced Treg levels were identified in preeclampsia, GDM, and pPROM as well as chorioamnionitis presenting with increased Treg dysfunctionality. Treg deficiencies may contribute to health issues in preterm newborns. Current sPTB treatments are limited, underscoring the potential of in utero therapies targeting inflammation, including T cell interventions. Future research aims to establish consensus on the role of Tregs in sPTB and associated conditions and advancing understanding of mechanisms leading to Treg deficiencies in adverse pregnancy outcomes.

## 1. Introduction

Spontaneous preterm birth (sPTB) is an unplanned, non-iatrogenic, live birth occurring before 37 weeks of gestation, where the newborn is not yet considered to be term and fully developed [1]. The negative impacts on both the health and growth of the baby have gained infamy as the leading cause of mortality and morbidity in neonates, responsible for the global death of nearly one million children under five [2]. sPTB can be classified into four gestational outcome categories, with 85% considered to be either ‘moderate’ with delivery occurring between 32 and 33 weeks or ‘late’ between 34 and 36 weeks, 10% are classified as ‘very’ preterm at 28 to 31 weeks, and the rarest cases, at 5%, are ‘extremely’ preterm, where delivery occurs 28 weeks or prior [3], with the poorest foetal outcomes associated with these earlier births. Prediction of sPTB and neonatal consequences remains difficult with most cases of sPTB devoid of a known cause, rendering them as idiopathic [4].

Several findings point to a link between the integrity of the maternal immune response and pregnancy outcomes [5,6,7], suggesting this could potentially hold the key to understanding which pregnancies make it to term while others deliver prematurely. With reduced regulatory T cell (Treg) presence and functionality reported in adverse pregnancy conditions, including sPTB [1], profiling and targeting this cell type in pregnancy holds promise to further our understanding and ability to predict idiopathic sPTBs. This review aims to explore the role of Tregs in a variety of conditions associated with sPTB to better understand their potential links to premature birth and drive further research to understand how Tregs may behave in idiopathic sPTB.

## 2. Influence of Tregs on Term Birth

Tregs are T cells broadly identified by the CD4 surface marker and occasionally CD8 [8]. Tregs are largely heterogeneous, with several subpopulations characterised by specific surface/internal markers. Each subpopulation of Tregs contributes a different role to the body’s immune response, but in general their function involves using immunosuppressive mechanisms to illicit an anti-inflammatory state [9]. Two common subpopulations of Tregs are the thymus-derived (tTregs) and the peripherally-derived (pTregs) [10,11]. Both subpopulations share common Treg markers, including CD25, GITR, CTLA4, and FOXP3, contributing to their role against both foreign and self-antigens [12]. Deficiencies in these expected markers can signify a disruption in immune response, with decreased FOXP3 showing increased rates of autoimmune and inflammatory diseases [13,14]. FOXP3, also known as the master transcription factor of regulatory T cells, is a forkhead box P3 protein that, when internally expressed in naive T cells, is key for their development into their regulatory form, allowing the identification of many Treg subsets. Although some subsets of regulatory T cells can be deficient in this marker, the majority of Tregs rely on FOXP3 as a key protein for their maturation and therefore their effective function through the promotion of anti-inflammatory states [15].

As demonstrated in Figure 1, Tregs implement anti-inflammatory states through mechanisms that lead to either the suppression of pro-inflammatory T effector cells (Teffs) (CD4+ CD25−) or the inhibition of antigen-presenting dendritic cells (DCs). These mechanisms can be regulated through the release of anti-inflammatory cytokines (i.e., interleukin-10 (IL-10) and transforming growth factor- β (TGF- β)), consumption of cytokines needed by Teffs, or the manipulation of biological pathways that can either trigger apoptosis or regulate T cell differentiation [16]. Alternatively, Tregs can suppress their target cells by cell:cell contact. For example, the co-inhibitory molecule PD-1 elicits its immunosuppressive functions through interactions with its ligand PDL-1 on effector T cells, resulting in exhaustion, neutralization, dysfunction, and increased IL-10 production by the Tregs themselves [17]. Additionally, the expression of CTLA-4 can also have immunosuppressive functions with its ability to cause trogocytosis of its ligands CD80/CD86 expressed on antigen presenting cells, leading to their immunosuppression [18].

These anti-inflammatory mechanisms allow for Tregs to induce antigen-specific tolerance, making them invaluable in pregnancy, which requires maternal immune tolerance of foetal antigens for successful placentation and development [20]. Maternal CD4+ CD25+ Tregs are present in both peripheral blood and the decidua basalis layer at the maternal-foetal interface [21,22]. Interestingly, approximately 95% of the Tregs present in the placenta in first trimester are classified as nTregs. Given that nTregs work against self-antigens and would theoretically be less effective in promoting foetal tolerance, it was suggested that the sheer number of nTregs and their ability to supress inflammatory T effector cells may play a bigger role than the presence of foetal-derived antigen-specific Tregs in the first trimester, but more research is needed to support this concept [23].

In healthy term pregnancies, the numbers of Tregs have been shown to change throughout gestation with a pro-inflammatory phenotype (less Tregs) characterising the uterus at the early stages of implantation. This is due to a high proportion of Th1 cells and cytokines such as TNF-α, IL-6, and IL-8 being present. Early inflammation is critical for blastocyst implantation into the uterine endometrium. This is corroborated in studies where TNF-α, a pro-inflammatory cytokine found in endometrial aspiration from IVF patients, is correlated with successful implantation [24]. However, as antigen presenting cells, such as dendritic cells, present small doses of foetal antigen to maternal Tregs, an anti-inflammatory is stimulated at the maternal-foetal interface, which is favourable for foetal tolerance and Treg expansion [23]. A dramatic expansion of Tregs shortly after implantation continues to increase until 20–30 weeks of gestation, which allows time for foetal growth and development. Towards the end of the third trimester, the number of activated Tregs decreases and this promotes another pro-inflammatory state, this time aiding in activation of parturition for a successful delivery [20]. It is suggested that these gestational changes are driven by fluctuations in hormones such as human chorionic gonadotropin (hCG), progesterone, and oestrogen, which can alter the expression of several immune cells present at the maternal-foetal interface [25,26,27].

## 3. Influence of Tregs on Spontaneous Preterm Birth

sPTB is a syndrome with several known and unknown causes, and this has resulted in difficulties in the early identification of pregnancies at risk and development of sPTB as well as the targeting of effective treatments. Most studies include participants with causes and risks of sPTB that are heterogenous in nature, which further complicates the picture [28]. This review includes several adverse pregnancy conditions and factors that are associated with an increased risk of sPTB: preterm premature rupture of membranes (pPROM), chorioamnionitis, bacertial and viral infections, dysbiosis of the cervico-vaginal microbiome, preeclampsia (PE), and gestational diabetes (GDM).

A key review [29], which holistically addressed the role of pregnancy immunology, highlighted an overall collaborative effort between cells of the innate (e.g., uterine natural killer (uNKs) and DCs) and adaptive (e.g., Tregs) immune responses that is crucial to avoid a premature trigger of sPTB. It was observed that reduced Treg suppressive functions in the third trimester could trigger labour of both term and preterm pregnancies, reaffirming the importance of the cell type beyond the first trimester due to its participation in the triggering of labour [29]. While Tregs present in labour of both spontaneous preterm and term births appear to be from the same subpopulations, it remains to be understood whether they occur as a result of birth or if their appearance is the mechanism driving the onset of labour, be that premature or term [30,31]. The number of Tregs and their suppressive function has been explored to identify characteristics of Tregs in sPTB, with some studies identifying decreased levels and impaired function of Tregs in sPTB compared to term births [32]. These findings are supported by mouse models, which have shown that depletion of Tregs in late pregnancy can result in preterm delivery [26] while Treg therapeutics support tolerance and a healthy term pregnancy through an improved maternal immune system [33]. This was demonstrated by Siddiq et al. [33] when lipopolysaccharide was administered to induce inflammation in the perinatal liver of mice was shown to be negatively regulated by Tregs.

In FOXP3+ Tregs specifically, a decrease in numbers were identified in comparison to their term birth counterparts. This is also supported by mouse models, which saw a link in the depletion of FOXP3+ Tregs and an increase in pregnancy failure [29]. Rowe et al. identified that pre-existing FOXP3+ Tregs retained from primary mice pregnancies drove Treg expansion in secondary pregnancies, indicating the importance of FOXP3+ Treg cells in immunological memory. Additionally, it was found that secondary pregnancies were more resilient to foetal reabsorption compared to primary pregnancies where FOXP3+ Treg cells were ablated. This highlights the ability of FOXP3+ Tregs to create lasting modifications to the maternal immune system, which contribute to successful pregnancies through a unique form of transferrable or sustained foetal tolerance [34].

### 3.1. Tregs in Adverse Pregnancy Outcomes Associated with PTB

With several complex causes driving sPTBs [35], it is essential to evaluate the commonalities between conditions that increase the risk of a premature birth. Although these conditions differ in their individual causes, they are tied by their association to premature labour and symptoms of inflammation. Better understanding of their inflammatory mechanisms and the role of Tregs in response to these different outcomes is key to discovering potential links between these mechanisms and why they may increase incidence of sPTB.

#### 3.1.1. Preterm Premature Rupture of Membranes (pPROM)

Preterm premature rupture of membranes (pPROM) is a pregnancy condition in which the foetal membranes rupture prior to 37 weeks’ gestation. It can be identified through the premature release of amniotic fluid [36]. Interestingly, pPROM has been found to precede about 40–50% of sPTB, yet the causes of pPROM and its association to other causes of sPTB remains unclear [37]. However, what is understood is that this rupture of membranes is strongly associated with infection and an inflammatory environment, showing that sPTB cases with pPROM have increased rates of infection/inflammation when compared to sPTB without pPROM [38]. One study found that 70% of pPROM cases had an additional diagnosis of intraamniotic infection, indicating the crucial role of these inflammatory states in pPROM and therefore sPTB [39].

The immunological implications of pPROM have been explored over the years, with several studies reporting cases of increased levels of pro-inflammatory cytokines present in the amniotic fluid, such as IL-8, IL-6, IL-1β, and tumour necrosis factor-α (TNF-α) [40]. This is supported by a recent study comparing the transcriptomic profiles of placental decidua and amniotic tissue from pregnancies with both pPROM and sPTB versus term pregnancies without pPROM. Tissues from the pPROM/sPTB group exhibited higher expression of pro-inflammatory Th1 and Th17 cells, while FOXP3, a characteristic internal marker of Tregs, was decreased in both chorionic and amniotic tissue [38,41]. Another study, performed on whole blood samples from pPROM mothers and their newborns, saw no significant difference in Treg levels when compared to non-pPROM pregnancies; however, the study was limited in that they did not use the intracellular staining of FOXP3 during characterisation. However, the study showed a higher presence of memory CD4+ T cells (CD45RA+, CD27−) and lower levels of naïve CD4+ T cells (CD45RA+) in the newborn whole blood. This led them to conclude that pPROM may trigger an early maturation of T cells during pregnancy, secondary to antigenic stimulation, playing a role in premature labour pathophysiology [42] This conclusion requires further explorations as early T cell maturation is suggested to be a broad consequence of sPTB, rather than viewing pPROM as a unique trigger [43].

#### 3.1.2. Chorioamnionitis

An example of a pro-inflammatory condition strongly affiliated with both pPROM and sPTB is chorioamnionitis, which is either caused by ascending infection via the cervix and/or vaginal tract into the uterus where it spreads to the chorioamnion of the placenta or through a cascade of proinflammatory signals that attract neutrophils [44,45]. This transmission can occur either with or without the presence of chorioamniotic membrane ruptures depending on the source of infection (bacterial, viral, fungal), with infective agents being present throughout the placenta and amniotic fluid [46,47]. Chorioamnionitis can cause inflammation and lesions in the placental membranes, along with maternal symptoms of infection including fever and increased white blood cell count. In the presence of chorioamniotic infection, pro-inflammatory chemokines and cytokines e.g., TNF-α, IL-6 etc., endotoxins, exotoxins, and granulocyte colony stimulating factors are released. This increase of inflammation within the uterus induces the influx of both maternal and foetal neutrophils to the maternal-fetal interface where lesions may be present [45,48]. Activated neutrophils at the membrane infection site release prostaglandins, which can cause cervical ripening and uterine contractility that promote preterm labour [49,50,51]. This is perpetuated by the presence of prostaglandins recruiting more neutrophils and causing production of matrix metalloproteinases and subsequent degradation of foetal membranes that can lead to a preterm labour and sPTB [48,51]. Studies have observed a diagnosis of acute chorioamnionitis in around 94% of preterm births occurring between 21–24 weeks gestation, compared to only 3–5% in term births [45,52].

Interestingly, a study exploring Treg levels and functionality in ‘moderate’ preterm birth and ‘late’ preterm birth with chorioamnionitis compared to term neonates without chorioamnionitis found that while the numbers of Tregs were not vastly different between the groups, Tregs from sPTB with severe chorioamnionitis neonates had low levels of functionality, which was assessed by the Tregs ability to suppress proinflammatory response from conventional T cells [53]. Their reduced suppressive function may provide an explanation for the manifestations of chorioamnionitis, given the Tregs would be unlikely to reduce inflammation caused by active pro-inflammatory lymphocytes at the maternal-foetal interface, such as Teffs and uNK cells, which would be responding to infective agents [54].

#### 3.1.3. Bacterial and Viral Infections During Pregnancy

A maternal immune system debilitated by an ongoing infection can also put the pregnancy at risk of premature delivery, with it being identified that 25–40% of preterm births may be attributed to intrauterine infections [55].

Bacteria are frequently identified in the amniotic fluid in cases of sPTB [51,56], specifically mycoplasmas, with the rate of sPTB being 14.3% higher when compared to mothers that had amniotic fluid negative for mycoplasma [57]. One current hypothesis linking bacterial infection to sPTB states that bacterial presence is identified and triggers stimulation of the developing foetal immune response, leading to a cascade of effects in utero. These events include the activation of toll like receptors (TLRs) that bind to specific epitopes on microorganism’s surfaces, specifically TLR-2 and TLR-4, which are highly expressed in the amniotic epithelium [58]. Premature uterine contractions may then occur after the release of pro-inflammatory cytokines such as IL-1β and TNF-α, resulting in the activation of neutrophiles and prostaglandin production [50].

Viral infections during pregnancy are also linked to a higher risk of sPTB, which is why understanding the cell-mediated responses of T cells in these cases become even more crucial [59]. One study assessing placental T cells isolated from mothers diagnosed with human immunodeficiency virus (HIV) identified an inverse CD4:CD8 T cell ratio in villous tissue between the HIV positive and HIV negative placental controls. This was attributed to a higher number of CD8+ T cells in the villi rather than lower levels of CD4+ T cells in cases of HIV positive pregnancies. This CD4:CD8 T cell ratio was further stratified by gestational age at delivery to observe any differences between the median gestational age for HIV positive cases (39 weeks’ gestation) and HIV negative cases (40 weeks’ gestation). Interestingly, they found no association between placental CD4:CD8 ratio between the gestational ages [60].

In the context of pregnancies where congenital human cytomegalovirus (HCMV) is diagnosed, the risk of having a sPTB was 15 times more likely than in pregnancies that were HCMV negative [61]. Circulating peripheral blood mononuclear cells (PBMCs), which encompass lymphocytes, monocytes, DCs, and NK cells, have been used to isolate Tregs, and one study doing so outside of pregnancy in HCMV positive individuals found their Tregs may decrease pro-inflammatory cells such as CD8+ T cells while increasing apoptosis of both CD8+ and CD4+ T cells [62]. However, when the relationship of Tregs and HCMV was explored in pregnancy results varied. One study infected placental extravillous trophoblasts (EVTs) with HCMV in vitro and observed their ability to increase the proportions of CD25HIFOXP3+ and PD1HI Tregs in comparison to non-infected EVTs. Overall, there was no differences between the Treg stimulation between both groups, but the authors hypothesised that this could be due to absent methods of localised induction of Tregs by EVTs or of antigen-presenting immune cells that is expected to be present at the maternal-foetal interface in vivo [15].

#### 3.1.4. Dysbiosis of the Cervico-Vaginal Microbiome

The cervical, vaginal, and endometrial microbiome function as a line of defence against several potentially dangerous bacteria that could enter vaginally during pregnancy and lead to several of the aforementioned conditions associated with sPTB. This protection is possible through the presence of anaerobic bacteria such as *Lactobacillus* spp., which produce lactic acid and create a highly acidic environment that aids in evasion of pathogens [57,63]. When studied in the context of non-pregnant mouse models with sexually transmitted diseases (STDs), vaginal *Lactobacillus* spp. have been shown to decrease the inflammatory response through the promotion of M2 macrophages and Tregs, which aided in wound repair at both the cervical and vaginal sites [64,65].

In another model, where non-pregnant mice were depleted of Tregs and infected with herpes simplex virus-2 (HSV-2), a delay in recruiting HSV-2 targeted CD4+ T cells was observed when compared to Treg sufficient mice. The authors suggested that this could be because Tregs in the vaginal mucosa, which are meant to be interacting at the site of the microbiome to recruit dendritic cells for CD4+ T cell activation, are absent [64]. Whether or not it can be concluded that Tregs have an important function within the cervico-vaginal microbiome remains unclear; however, a recent study looking at Tregs in non-pregnant women following clinical diagnosis with human papilloma virus (HPV) found inflammation caused by HPV dysregulating the cervico-vaginal microbiome was associated with higher levels of circulating Tregs (CD4+ CD25+ FOXP3+) within the peripheral blood [66].

The maintenance of a regulated cervico-vaginal microbiome and interacting cells is crucial to the prevention of sPTB, with several studies showing instability of the microbiome to be a leading risk factor for premature labour [63,64,67]. Many of the findings focus on the importance of *Lactobacillus* species, with women whose microbiome showed *Lactobacillus* was not the predominant bacterial species being at an increased risk of sPTB compared to healthy pregnant microbiomes where *Lactobacillus* was abundant with diverse species strains [60,68,69]. However, it should be noted that several of these studies did not represent a diverse sample population or account for potential confounding variables that influence the cervico-vaginal microbiome, including ethnicity, body mass index (BMI), or menstrual cycle phase.

#### 3.1.5. Preeclampsia (PE)

Preeclampsia (PE) is a condition occurring in pregnancy, typically diagnosed around 20 weeks’ gestation, characterised by the development of hypertension (≥140/90 mmHg) and evidence of maternal organ dysfunction (i.e., renal, hepatic, and haematological dysfunction). Proteinuria, which is often a sign of kidney damage, used to be involved in classifying PE, but it is no longer regarded, given that the condition can manifest in organ issues beyond kidney dysfunction [70]. Maternal obesity, older maternal age, and having hypertensive-related diseases in previous pregnancies are a few of the many risk factors that can result in PE [4]. It is a condition with grave consequences, which include maternal mortality and foetal growth restriction, which is why medically-induced preterm births are usually initiated [71].

Decreased Treg cell numbers are associated with preeclampsia, with several studies exploring their role in the maternal immune response in hypertensive pregnancies [4,20,72,73,74]. In Sasaki et al. [73], the number of Treg cells was compared between healthy pregnant women and non-pregnant women to those with pre-eclamptic pregnancies, and a disparity was found. Those with pre-eclampsia were found to have fewer Tregs, specifically CD25 bright T cells, whereas CD8+ T cells in the placental bed biopsies were found to be higher in the PE patients than the healthy pregnant women. FOXP3 expression was located on most CD4+ CD25 bright Treg cells in all groups; however, in the placental bed biopsies, FOXP3+ Treg cells were also reduced in the pre-eclampsia group [73].

Lower numbers of FOXP3+ CD4+ CD25 bright Tregs was most commonly found in the maternal peripheral blood and placental bed biopsies of PE patients as opposed to the non-pregnant or healthy pregnant women. This suggests that there is a correlation between low Treg numbers and preeclampsia that can eventually require a medically induced preterm birth [73]. This was confirmed in results by Prins et al. [74], who identified reduced levels of CD4+ FOXP3+ Tregs from isolated peripheral blood from pregnant women with PE when compared to healthy pregnant controls.

Moreover, the meta-analysis published by Green et al. [4] further highlights this correlation between PE and Treg levels during pregnancy. Out of the 32 studies that compared the number of Tregs in women with preeclampsia and healthy pregnancies, 30 found significantly higher Treg levels in the peripheral blood of healthy pregnant individuals. An association between Tregs levels and PE is strongly supported in the literature, but further characterisation of what Tregs manifest in different PE phenotypes is crucial to improving outcomes [4] and understanding associations between preeclampsia and preterm birth.

#### 3.1.6. Gestational Diabetes (GDM)

Gestational diabetes mellitus (GDM) is one of the most common endocrinopathies that occurs during pregnancy and is characterised by hyperglycaemia and glucose intolerance. Associated maternal complications that can arise from this condition include pre-eclampsia and type 2 diabetes, which manifests post-natally. For the foetus, GDM can result in intrauterine death and foetal malformation [75]. Additionally, newborns can also experience hypoglycaemia and macrosomia, with the additional risk of type 2 diabetes during their adult life [76]. It is currently thought that the inability of the mother’s immune system to adapt to pregnancy may lead to immune dysregulation, which is a key proponent in the development of insulin resistance and chronic inflammation seen in GDM [77,78].

As previously mentioned, Treg involvement varies throughout a healthy gestation, with a pro-inflammatory phenotype characterising early implantation [79] that is then followed by Treg expansion, followed by a steep decline in the number of Tregs prior to the onset of labour [80]. It is currently thought that if the maternal immune system does not follow these expected patterns, there may be underlying immune dysregulation. This dysregulation is a key proponent in the development of insulin resistance and chronic inflammation seen in GDM, characterised by factors such as increased levels of pro-inflammatory markers TNF-α and IL-6 [81,82].

To understand the specific potential pathogenic role that Tregs have in GDM, a study by Schober et al. [76] investigated disparities in Treg subsets and their suppressive capabilities across healthy pregnant women, pregnant women with insulin dependent GDM, and pregnant women with dietary adjusted GDM. There was a focus on assessing the CD4+ CD127low+/− CD25+ FOXP3+ Tregs, which in this study was further divided into naïve CD45RA+ Tregs, HLADR-CD45RA- memory Tregs (DR-Tregs), HLADRlow+ CD45RA- memory Tregs (DRlow+ Tregs), and HLADRhigh+ CD45RA- memory Tregs (DRhigh+ Tregs) [76].

To analyse these cells, peripheral blood samples were taken from pregnant women of each group, and flow cytometry was then performed. Similar to findings in preeclampsia, decreased Treg levels were characteristic in the GDM pregnant women when compared to non-GDM cohorts, particularly naïve CD45RA+ Tregs subpopulations. This study supports various others that found reduced Tregs linked to adverse pregnancy outcomes [83,84,85].

A metanalysis by Arain et al. [86] studied the Treg levels in the mother’s peripheral blood and its correspondence to developing GDM. Women with GDM were looked at alongside pregnant women without GDM. In several of the studies, analysed Treg levels were lower in the GDM cohort. It can be inferred that GDM has immune implications that manifest in ways in that cause low Treg cell counts in the maternal peripheral blood [86].

## 4. Influence of Tregs on Preterm Neonatal Outcomes

In parallel with interrogating the role of Tregs in pregnancy and at the maternal-foetal interface, there is also a need to determine how these in utero exposures influence immune development and morbidity risk in preterm newborns. It is known that maternal response to infection and inflammation alters the functionality of the neonatal immune system [87], with lots of research focusing on the impact this has on newborn immune function and health outcomes [88,89,90].

### 4.1. Sepsis

According to Kamdar et al., both term and preterm newborns follow a similar trajectory in postnatal immune system development, yet sPTB infants were still at a higher risk for developing sepsis when compared to their term counterparts [91]. This may be due to decreased function of T cells, given that findings show a decreased ability of preterm T cells to produce IL-8, which is important for attracting neutrophils to combat infection [91]. This study also used flow cytometry to compare longitudinal PBMCs from preterm newborns with either no suspicion of infection following antibiotics, suspicion of infection/sepsis and given further course of antibiotic treatment, or if confirmed with chorioamnionitis. Results showed that the infants with suspected sepsis or who had chorioamnionitis had a lowerFOXP3 expression on their Tregs compared to those without infection, suggesting preterm newborns may see lower levels of this cell than their preterm counterparts without suspected/confirmed infections [91].

The involvement of Tregs in two major neonatal conditions associated with preterm birth, necrotising enterocolitis and neonatal encephalopathies in premature infants, highlight the importance of further research as our understanding of the roles that Tregs mechanisms play is still limited.

### 4.2. Necrotising Enterocolitis (NEC)

Necrotising enterocolitis (NEC) is an inflammatory condition commonly developed by preterm newborns that affects the intestine to a degree that can be life-threatening and require emergency treatment [92]. NEC morbidity and mortality rates are the highest in preterm newborns since they are quite fragile and have greater difficulty recovering despite surgical advancements and other intervention strategies [93,94]. With NEC, the neonatal immune system is believed to have issues regulating present inflammation, especially in cases of sPTB, with babies born prematurely found to have decreased numbers of Tregs and being unable to maintain the integrity of the intestinal epithelium [95]. To explore this concept, one study took peripheral blood samples from newborns diagnosed with NEC to compare with those without NEC, using flow cytometry to identify their respective presence of circulating Tregs. Findings identified lower numbers of Tregs within the peripheral blood of those with NEC compared to controls. This study recognised that Treg proliferation may be defective in cases where NEC is present, finding a reduction in FOXP3 positive Treg expression. Like many other conditions associated with preterm birth, additional research is required to explore a solid causation for this pathological association [95].

These findings are interesting when considered in parallel to studies that found neonates play an active role in immune defence through observation of T cells in the amniotic fluid that express surface markers (i.e., CD103 and CD161), indicating a potential mucosal origin site for the cell. It would be beneficial to explore if preterm babies develop NEC due to insufficient Tregs, therefore being unable to repair the intestinal barrier, or if there is a lack of Tregs due to NEC damaging the foetal intestine, where foetal T cells may find their origin [41].

### 4.3. Encephalopathies of Premature Newborns

Encephalopathies in premature infants are commonly reported neurological conditions that impact several parts of the developing newborns brain, including the cerebellum and the white and grey matter. Encephalopathies following sPTB often lead to long term health consequences that can impair vision, decrease motor coordination, delay cognitive development, and much more besides. Encephalopathy of prematurity can be characterised by using magnetic resonance imaging (MRI) to observe myelin sheath reduction, immature oligodendrocytes, lesions of the axon, and neuroinflammation [96]. Gestational age has been explored in the context of neonatal encephalopathy (NE), noting that neonates born between 35–38 weeks’ or after 40 weeks’ gestation saw an increased risk of NE compared to those born from 39–40 weeks, indicating potential significance for late sPTBs occurring from 35–37 weeks GA developing encephalopathy. The risk was further increased if the mother had an infection or reported autoimmune disease, specifically rheumatoid arthritis [97].

The presence of neuroinflammatory mechanisms being involved in a condition commonly reported in premature infants has prompted the exploration of lymphocytes and their role in neonates developing NE/EOP. One review noted two different studies conducted on murine models with induced hypoxic ischemia, where in one case treatment with FTY720 (a T cell migration blocking agonist receptor) depleted Th17 T helper cells saw a decrease in pro-inflammatory cytokines and preservation of white cerebral matter [98]. However, the other study, where FTY720 depletion treatment was performed on Tregs, CD4+ T cells, and CD8+ T cells, saw a loss of both white and grey cerebral matter in newborn pups [99]. The comparison between these studies is limited since they depleted different T cell types; however, it is an interesting foundation for exploring T cell mechanisms in NE.

A more recent study compared the presence and functionality of lymphocytes in the peripheral blood of school-aged children who experienced NE as neonates with that of children with cerebral palsy and age matched healthy controls. Results showed that NE neonates and their healthy counterparts had similar values of circulating T cells, but levels of effector T cells, which produce pro-inflammatory cytokines, were significantly higher in the NE infants. They stated that further research specifically exploring Treg numbers and several T helper subpopulations would be needed to determine their respective roles in NE. However, they suspect their findings to be evidence of the role played by lymphocytes, such as various T cells, in the disease mechanism [100].

### 4.4. Bronchopulmonary Dysplasia (BPD)

Another condition that often occurs in preterm neonates includes bronchopulmonary dysplasia (BPD), which is a chronic lung disease developed from injury to the evolving lung tissue [101,102]. Supplemental oxygen, positive pressure ventilation, and sepsis occurring after birth are all common causes of the condition and are linked to inflammatory processes [102]. BPD’s link to inflammation is strengthened by the studies identifying affected neonates as more inflammatory signals and increased cytokine/chemokine expression, meant to promote inflammation of the alveolar epithelial cells [102,103,104].

This has made the anti-inflammatory role of Tregs a recent point of interest within BPD studies, where findings conflict on Treg numbers and phenotype within the condition. When investigating the amount of Tregs in umbilical cord blood samples of preterm neonates, Misra et al. found that Treg numbers were significantly decreased when they had cases of moderate BPD compared to mild or absent cases of the condition [105]. Findings of low levels of anti-inflammatory cells and an increase in pro-inflammatory cytokines, such as IL-6 in humans [103,105], fit the understanding that BPD is associated with inflammation and provide insight into specific pathways that may be driving this condition so closely linked to sPTB. A recent study wanted to explore the protective role Tregs may play in a BPD mouse model by knocking out the Treg co-transcription factor interferon regulatory factor 4 (IRF4), which is involved in pro-inflammatory processes [104]. They found that elevated levels of IRF4 led to a decrease in FOXP3 expression and Treg numbers, and that knocking down IRF4 could increase anti-inflammatory states to improve lung development [104]. For the purpose of this review, this study is limited in that it did not explore effects of the knockout on gestational age. Despite this, it is an important direction for future research.

Contrary to most studies discussing a decrease in Tregs, a 2020 study by Pagel et al. looked at the peripheral blood of preterm neonates and found that Tregs were increased in number and function in the first two weeks before BPD was developed compared to term and adult blood [106]. They specifically observed naïve Tregs (CD45RA+/HLA-DR-/Helios+) as being increased in the time following birth, then observed a higher population of activated Tregs (which included markers such as CCR6+, HLA-DR+, and Ki-67+) in the week following delivery [106]. This highlights the importance of identifying Treg subpopulations in preterm neonates for a better understanding of disease pathophysiology and treatments. The study addresses the contradictions of their results to what has widely been observed in other sPTB neonates with BPD by noting that the study did not account for confounding variables like gestational age and presence of sepsis, as well as having differing methodological approaches to the other studies conducted [106].

### 4.5. Intracerebral Haemorrhage (ICH)

sPTB leaves newborns in a very vulnerable state, as several of their organs may remain underdeveloped depending on the gestational age at birth. Preterm newborns may have fragile blood vessels that can leave them prone to bleeds and strokes [107]. Intraventricular haemorrhages (IVH) are a type of intracerebral haemorrhage (ICH) where blood leaks into ventricles and fluid-filled areas around the brain, affecting a disproportionate amount of preterm compared to term neonates [108]. In the case of IVH in sPTB, where the germinal matrix vasculature is especially fragile, there is an increase in pro-inflammatory cytokines and proteins such as IL-6, IL-1β, and TNF that can lead to sustained cerebral inflammation [109,110,111].

Outside of the context of pregnancy, Tregs have been suggested to play a role in promoting immune homeostasis after an ICH in both rodent models and human participant samples [112,113,114]. However, their link to preterm neonates remains understudied. The mechanisms by which Tregs manage inflammation for ICH in preterm neonates hold potential for further investigation given ICH’s ties to sPTB and the elusive role Tregs may play in the condition [109].

## 5. Potential Immunotherapies for Adverse Pregnancy Outcomes Associated with sPTB

The involvement Tregs have shown in adverse pregnancy outcomes associated with sPTB and poor neonatal health consequences have led to a push for studies to explore their role in the prevention of premature labour. Treg therapies have been used in other inflammatory conditions, but whether these therapies have a place in the context of sPTB remain unclear yet possible.

### 5.1. Treg Cell Infusion Therapy

Treg cell therapy is proving useful for several immune mediated diseases and beyond, looking to address the issue of a dysfunctional immune response, that can lead to excessive inflammation in several conditions. There are a variety of approaches currently being assessed in both preclinical and clinical trials that have different benefits that should be considered based on the type of condition the Tregs are being used for. Each of these approaches begin with Tregs isolated from peripheral blood and expanded in the presence of IL-2. One of the most common approaches includes polyclonal expansion of Tregs using anti-CD3/CD28 beads, allowing for a larger cell yield and potency at the expense of specificity. Another method, used primarily for cases in solid organ transplantation, involves presenting the IL-2 activated Tregs with antigen presenting cells from the donor to stimulate an antigen-specific Treg response. While these are more specific than the polyclonal Treg method, they usually do not have a high enough cell yield, even following cellular expansion. The third most common Treg product used as a cell therapeutic involves the expansion of polyclonal Tregs that are then transduced with either a chimeric antigen receptor (CAR) or an artificial T cell receptor (TCR), the point behind each to target a specific antigen, the latter of which involves specific major histocompatibility complex (MHC)-peptide complexes. The benefit of this method is that it allows for a high cell yield, specificity, and potency [115,116].

A study by Gomez-Lopez et al. tested the potential of Treg infusion therapy for preterm birth using a mouse model with either partial or total depletion of Tregs at three weeks gestation [26], which is comparative to the late third trimester in human pregnancies [117]. Depletion was performed by injecting FOXP3DTR mice with various amounts of diphtheria toxin to deplete FOXP3+ Tregs prior to mating with BALB/c mice. As a result, 15% of the mice produced premature pups, which were reported as smaller and leaner than the mice without Treg depletion. An injection of Tregs was then applied to the mice with complete FOXP3+ Treg depletion to see if this prevented preterm birth in pregnancy. This therapy was able to significantly decline the rate of preterm births from 15% to 0%, and the pups born after Treg infusion had a greater survival rate compared to the group withheld [26].

This evolving therapy shows potential clinical benefits for pregnant women in increasing their Treg levels and preventing sPTB. To further test that this form of treatment is transferrable to humans, Dudreuilh et al. [118] experimented with transferring Tregs into human leukocyte antigen (HLA) sensitised patients in an ongoing trial. It is possible for some kidney failure patients to become sensitised to HLA, preventing their body from accepting a kidney transplant, so the intention behind this study is to test the ability of Tregs in these patients to determine whether memory B and T cells can suppress HLA to facilitate their kidney transplant. If these findings show improvements in tolerance in instances of organ transplant, that opens the door to Treg cell therapy for tolerance of foetal antigen by the maternal immune system.

### 5.2. Probiotics for the Prevention of sPTB

Probiotics have been used throughout history as a treatment for several ailments, particularly those related to intestinal disorders, given that they illicit microbial and immune interactions that combat inflammation [119]. Studies looking at the mechanisms used by probiotics to incite these anti-inflammatory processes show that different strains of probiotics can lead to the production of cytokines such as IL-12, which can stimulate Th1 and NK cells for a pro-inflammatory immune response. However, immunomodulating probiotic strains have been shown to stimulate Tregs and anti-inflammatory cytokine IL-10. These findings indicate the importance of understanding exactly which strains of probiotics should be used, with *Lactobacillus* spp. stimulating either pro- or anti-inflammatory mechanisms [120]

In the context of human pregnancy, probiotics have primarily been explored to prevent neonatal symptoms associated with a preterm birth, such as cases of NEC and low birth weight. When studies explored the effects that probiotics administered in pregnancy would have on breast milk, an increase in anti-inflammatory responses from IgA and TGF-β1 in breast milk and IFN-γ in cord blood were elicited [121]. More studies are needed to confirm the use of probiotics as a preventative measure for preterm births as most of the studies in humans have restricted sample sizes and differ in the bacterial strains used as well as duration of probiotic administration. These limitations make it difficult to draw substantial conclusions regardless of the studies showing benefits in reducing inflammation and aiding in sPTB neonatal outcomes [122,123].

### 5.3. Tocilizumab Use for Inflammatory Rheumatoid Arthritis

Rheumatoid arthritis (RA) in adults and juvenile idiopathic arthritis (JIA) are chronic autoimmune inflammatory joint diseases from which pain and stiffness are encountered. Therefore, due to the nature of these diseases, immune dysregulation is heavily implicated in their pathogenesis. This has been demonstrated in a meta-analysis by Falcon et al. where pregnant women with RA were identified to have reduced functionality of CD4+ CD25high regulatory T cells and increased pro-inflammatory cytokines IL-6, IFN-(gamma), and TNF-α in maternal blood collected during pregnancy in the third trimester [124].

It has been found that pregnant women with these types of inflammatory arthritis are at a greater risk of having preterm deliveries than their counterparts without arthritis [125]. Further testing was conducted by Smith et al., who assessed the risk of sPTB for pregnant women who had either RA or JIA and compared them to pregnant women without arthritis. In both groups of arthritis women combined, they had a much higher risk of going into preterm labour, specifically a moderate sPTB at around 32 weeks of gestation [126]. For individuals who were diagnosed specifically with RA, they had an identified increase in extreme sPTBs compared to the control and even JIA group [126].

Immunomodulatory drugs are emerging to enhance Treg levels in patients with different autoimmune diseases, including RA. Tocilizumab is an example of one such pharmaceutical, which acts as an anti-interleukin-6 receptor antibody, that can be used to limit the inflammation seen in RA patients [127]. Kikuchi et al. carried out a study where non-pregnant RA patients were given Tocilizumab and then their peripheral blood cells were assessed by flow cytometry. Based on the clinical disease activity index (CDAI), it was determined that 53.8% of RA patients went into remission as their CD4+ CD25+ CD127low Tregs and HLA-DR+ activated Tregs increased, resulting in a decrease in clinical symptoms of inflammation [127]. The use of Tocilizumab increased Tregs, improving clinical symptoms of inflammation in RA patients. Even though the mechanism by which low Tregs and RA operate has yet to be fully understood, Tocilizumab is still able to raise Treg levels via inhibiting the IL-6 pathway and increasing immunosuppression in these patients [127,128].

Interestingly, in a mouse model using Tocilizumab as a preventative intervention for preterm delivery, it was found that the drug could reduce LPS-induced inflammation and prolong gestation [128]. Whether this is translational to human pregnancies remains to be explored, but from the studies that have been conducted thus far there has been found to at least be no reportable increase in spontaneous abortion or congenital issues [129,130]. It is believed that Tocilizumab does not cross the placenta in high amounts during the first trimester when most congenital defects may begin to form. However, in later trimesters, there have been increased incidences of sPTB, but whether this is due to the drug or the individual having RA is unclear [131]. More research is needed looking at not only the safety of Tocilizumab in pregnancy but also its ability to increase Tregs and how this affects rates of sPTB in order to identify if it would be a viable treatment for at risk pregnancies.

## 6. Limitations

The limitations of this literature review comprise the inclusion of studies that use murine models as the basis for their evidence and thus this highlights the need for increased human models in the research of the Tregs in adverse pregnancy outcomes associated with sPTB. This would increase the validation of the findings and allow greater understanding of the cellular processes discussed in human biology. Moreover, small sample sizes and varying methodologies limit the conclusions that can be made from the research. Improvements in these limitations would make room for standardised research, allowing the opening of avenues to explore preventative and curable treatment options for the pathologies discussed.

## 7. Discussion

As a leading cause of neonatal morbidity and mortality, preterm birth remains a pervasive issue in the context of perinatal health. sPTB is often accompanied by immune dysregulation and heightened inflammatory responses, indicating an effected immune response within conditions affiliated with increased risk of premature labour. This review set out to explore the relationship between Tregs and various pregnancy conditions that may influence sPTB occurrence, examining how Treg levels and functionality impact sPTB phenotypes and associated neonatal outcomes.

The literature consistently points to a variation in Treg levels and functionality in the context of these inflammatory pregnancy conditions. Some studies indicate a decrease in Treg levels, while others suggest Treg dysfunctionality rather than a numerical decrease. In sPTB cases specifically, findings show increased presence of pro-inflammatory cytokines, with some subpopulations of Tregs, such as memory Tregs, being elevated. Tregs also make frequent occurrences in the literature, with their decreased levels/functionality being suggested as an important marker of premature labour. The variations seen among Treg findings underline the complexity of the immune landscape in sPTB, highlighting the need for standardised research to fully understand the impact of Tregs on different sPTB phenotypes.

Tregs have been implicated in conditions affiliated with sPTB such as neonatal sepsis, necrotising enterocolitis, and others, where their level of functionality has been proposed to contribute to dysregulated immune response and long-term effects on neonatal health. There is still much to be understood in these areas due to the current limitations, including small sample sizes and varying methodologies, which necessitate further research to confirm these findings. Additional studies regarding T cell pathways should be explored for better understanding of how Tregs can modulate inflammation in these cases and protect against long term health consequences in premature newborns.

Given these insights, Tregs may prove fruitful in guiding in utero therapeutics to decrease inflammation and potentially prevent sPTB. Treatments aimed at enhancing Treg functionality or levels, such as Tocilizumab in RA patients, have shown promise in reducing inflammation and improving clinical outcomes. Tocilizumab increases Treg levels and reduces inflammatory markers, potentially mitigating the risk of preterm birth in RA patients. Other therapeutic strategies, such as Treg therapy, have shown success in animal models, where Treg depletion increased sPTB risk and Treg administration reduced risk, improving neonatal outcomes. Probiotic administration has also demonstrated potential by increasing Treg levels and reducing pro-inflammatory Th17 cells in both preterm and full-term mice. Whether these treatments would benefit pregnant humans at risk for sPTB remains to be tested but there is great potential in their use to decrease inflammation and create the possibility for longer gestations in pregnancies at higher risk of sPTB.

In conclusion, while the relationship between Tregs and sPTB is complex and multifaceted, the findings of pregnancy outcomes associated with sPTB suggest that enhancing Treg functionality and levels could potentially reduce sPTB risk and improve pro-inflammatory states often seen in adverse preterm neonatal outcomes. Future research should focus on elucidating the specific mechanisms utilised by Tregs in idiopathic sPTB. This could be explored by identifying potential links between Treg numbers and functionality in idiopathic sPTB and those suggested in associated adverse pregnancy conditions. Similarly, further work developing targeted therapies to modulate Treg activity in utero should be conducted, potentially transforming the management and prevention of sPTB associated complications.

## Figures and Tables

**Figure 1 ijms-25-11878-f001:**
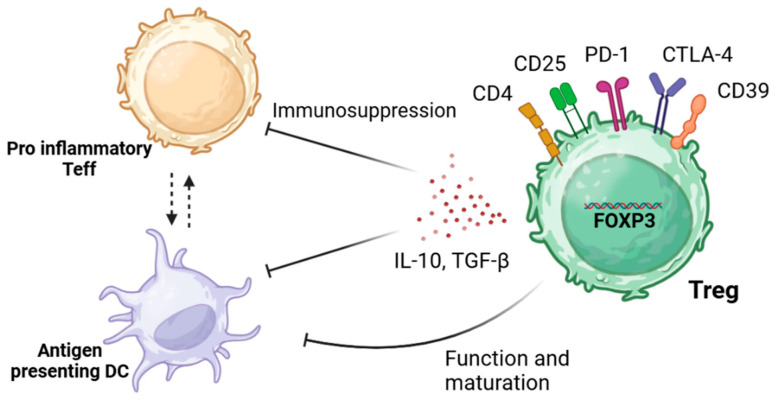
Suppressive mechanisms elicited by Tregs. Tregs can elicit their immunosuppressive functions through various mechanisms. These include the expression of co-inhibitory external markers such as PD-1 or CTLA-4 through cell:cell contact and the production of anti-inflammatory cytokines IL-10 and TGF-β. This allows for the inhibition of the function and maturation of DC and overall immunosuppression of pro-inflammatory effector T cells (Teffs). This image was created in BioRender, adapted from Kempkes, R. W. M., Joosten, I., Koenen, H. J. P. M., and He, X., 2019 [19].

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
