# Peer review of "The Immunomodulatory Role of Regulatory T Cells in Preterm Birth and Associated Pregnancy Outcomes"

_ijms, 2024, doi:10.3390/ijms252211878_

Round 1

Reviewer 1 Report

Comments and Suggestions for Authors

In the narrative review entitled “The immunomodulatory role of regulatory T cells in preterm birth and associated pregnancy outcomes”, the authors provide a thorough analysis on the impact of regulatory T cells in preterm birth and neonatal outcomes. Overall, it is a well written and comprehensive review.

My minor comments are:

-              Abstract should include the key results of the review.

-              The authors should include a limitations section.

-              The authors should elaborate more on future directions and clinical implications of their study.

Author Response

  1. Abstract should include the key results of the review. Thank you for your comment we have addressed these results in lines 26-31.
  2. The authors should include a limitations section. Thank you for your comment. We have added an additional limitation section in lines 644-653 that aims to address this.
  3. The authors should elaborate more on future directions and clinical implications of their study. Thank you for your comment, we have elaborated and made the future directions and clinical implications clearer in 681-703.

Reviewer 2 Report

Comments and Suggestions for Authors

This article presents a literature review of the current knowledge on the potential roles of regulatory T cells (Tregs) in the pathogenesis of preterm birth (PTB), focusing on comparing Treg presence and function in pregnancies resulting in PTB in comparison with healthy term pregnancies.

Given that PTB is the leading cause of infant mortality and morbidity, significantly increasing the risk of chronic diseases later in life, and considering the incomplete understanding of its underlying causes and mechanisms, the topic is of substantial interest.

However, the paper requires major improvements, particularly in the organization of chapters and in providing a more thorough analysis and discussion of the presented results.

Overall Suggestions for Improvement: The chapter organization could be more logical and comprehensive, with suggested sections structured as follows:

  1. Introduction
  2. Influence of Tregs on Term Birth
  3. Influence of Tregs on Preterm Birth

3.1. Tregs in Adverse Pregnancy Outcomes Associated with PTB

3.1.1. Preterm Premature Rupture of Membranes (pPROM)
3.1.2. Chorioamnionitis
3.1.3. Bacterial and Viral Infections during pregancy
3.1.4. Dysbiosis of the Cervico-vaginal Microbiome
3.1.5. Preeclampsia (PE)
3.1.6. Gestational Diabetes Mellitus (GDM)

3.2. Tregs in Idiopathic PTB

  1. Influence of Tregs on Preterm Neonatal Outcomes

4.1. Sepsis

4.2. Necrotising Enterocolitis (NEC)

4.3. Encephalopathies of premature newborns

4.4. Bronchopulmonary Dysplasia

4.5. Intracerebral Hemorrhage

4.6.

  1. Potential Immunotherapies for adverse pregnancy outcomes associated
  2. Discussion

Detailed Proposed Changes:

Graphical Abstract: Adapt the graphical abstract to include all mentioned neonatal outcomes (notably, the microbiome is missing from the current visual representation).

1. Introduction: The authors should clearly differentiate between idiopathic sPTB and medically indicated PTB and decide which one to focus on throughout the paper. Is the goal to explore the influence of Tregs on idiopathic sPTB, or to elucidate their role in mechanisms related to known causes of PTB? While the introduction frames the issue of idiopathic PTB, the later sections address medically indicated states, such as infections, that lead to PTB. This distinction should be clarified early in the introduction.

2. Influence of Tregs on Term Birth:

    • Line 79: Clarify what "these" refers to in the sentence.
    • Line 86: Include cell-cell interactions in Figure 1 for clarity.
    • Lines 111-122: This section requires more in-depth explanation. Additionally, it would be helpful to include a visual representation of different Treg expressions in healthy pregnancies for better understanding.
    • Line 125: Rename the subsection "Tregs in Spontaneous Preterm Birth" to “3. Influence of Tregs on Preterm Birth” to maintain consistency with the new chapter structure.
    • Line 130: Specify the adverse pregnancy conditions and factors mentioned in the text.
    • Line 132: When referring to the "key review," provide the appropriate citation.
    • To improve readability, organize the results discussed across this chapter in a table to emphasize key findings.

3. Inflammatory and Immunocompromising Pregnancy Outcomes Associated with sPTB:

    • Rename the current section "Inflammatory and Immunocompromising Pregnancy Outcomes Associated with sPTB" to "3.1. Tregs in Adverse Pregnancy Outcomes Associated with PTB" to align with the chapter reorganization.
    • Line 171: Change the level of this section to 3.1.1, titled "Preterm Premature Rupture of Membranes (pPROM)."
    • Lines 188-195: This section requires more detail. Additionally, the authors should discuss further research to confirm whether pPROM triggers early T cell maturation during pregnancy.
    • Line 256: Provide a clear explanation of PBMCs.

4. Influence of Tregs on Preterm Neonatal Outcomes:

    • Line 380: Add a subtitle to introduce this section more clearly (e.g., 4.1 Sepsis).
    • Lines 392-395: The section would benefit from the inclusion of more neonatal conditions affected by Tregs, such as bronchopulmonary dysplasia and intracerebral hemorrhage, as they can also be influenced by Tregs.

6. Discussion: The authors need to ensure consistency throughout the manuscript regarding the objective of the review. Is the focus on the relationship between Tregs and various pregnancy conditions that may influence and lead to sPTB, or, as stated in the title, is the review aimed at exploring the roles of Tregs in spontaneous PTB itself, and how they may contribute to different pregnancy outcomes? This should be clearly articulated in the discussion and aligned with the rest of the manuscript.

Author Response

This article presents a literature review of the current knowledge on the potential roles of regulatory T cells (Tregs) in the pathogenesis of preterm birth (PTB), focusing on comparing Treg presence and function in pregnancies resulting in PTB in comparison with healthy term pregnancies.

Given that PTB is the leading cause of infant mortality and morbidity, significantly increasing the risk of chronic diseases later in life, and considering the incomplete understanding of its underlying causes and mechanisms, the topic is of substantial interest.

However, the paper requires major improvements, particularly in the organization of chapters and in providing a more thorough analysis and discussion of the presented results.

Overall Suggestions for Improvement: The chapter organization could be more logical and comprehensive, with suggested sections structured as follows:

  1. Introduction
  2. Influence of Tregs on Term Birth
  3. Influence of Tregs on Preterm Birth

3.1. Tregs in Adverse Pregnancy Outcomes Associated with PTB

3.1.1. Preterm Premature Rupture of Membranes (pPROM)
3.1.2. Chorioamnionitis
3.1.3. Bacterial and Viral Infections during pregancy
3.1.4. Dysbiosis of the Cervico-vaginal Microbiome
3.1.5. Preeclampsia (PE)
3.1.6. Gestational Diabetes Mellitus (GDM)

3.2. Tregs in Idiopathic PTB

  1. Influence of Tregs on Preterm Neonatal Outcomes

4.1. Sepsis

4.2. Necrotising Enterocolitis (NEC)

4.3. Encephalopathies of premature newborns

4.4. Bronchopulmonary Dysplasia

4.5. Intracerebral Hemorrhage

4.6. …

  1. Potential Immunotherapies for adverse pregnancy outcomes associated
  2. Discussion

Thank you for your comment, we have made this change and since clarifying the focus of the paper to better understand the role Tregs play in conditions associated with sPTB, so as future research may be able to tie this into understanding the causes of idiopathic sPTB we have taken out the heading ‘Tregs in idiopathic PTB’.

Detailed Proposed Changes:

Graphical Abstract: Adapt the graphical abstract to include all mentioned neonatal outcomes (notably, the microbiome is missing from the current visual representation). Thank you for your comment, the additional neonatal outcomes have been added as well as the section discussing Tregs in dysfunctional microbiome.

  1. Introduction:The authors should clearly differentiate between idiopathic sPTB and medically indicated PTB and decide which one to focus on throughout the paper. Is the goal to explore the influence of Tregs on idiopathic sPTB, or to elucidate their role in mechanisms related to known causes of PTB? While the introduction frames the issue of idiopathic PTB, the later sections address medically indicated states, such as infections, that lead to PTB. This distinction should be clarified early in the introduction.

Thank you for your comment, we have addressed this in lines 66-68 by clarifying that our focus is on what has been investigated surrounding Tregs in conditions associated with sPTB. In doing so we hope that future research may be able to explore potential links between these conditions and cases of idiopathic sPTB to help understand their potentially shared mechanisms and potential predictors/therapeutics that can be utilised in a clinical setting.

  1. Influence of Tregs on Term Birth:
    • Line 79: Clarify what "these" refers to in the sentence.- We agree you’re your comment and have since changed to ‘Tregs implement anti-inflammatory states’ in lines 84-85.
    • Line 86: Include cell-cell interactions in Figure 1 for clarity. Thank you for your comment, we have included the cell: cell interactions present to improve clarity in lines 92-93.
    • Lines 111-122: This section requires more in-depth explanation. Additionally, it would be helpful to include a visual representation of different Treg expressions in healthy pregnancies for better understanding. Thank you for your comment, we have included more detail in text to provide further information on the expected change of Tregs in healthy pregnancies. The processes detailed allow for improved understanding of how they allow for fetal tolerance and promote an anti-inflammatory state until the end of pregnancy nears and a pro-inflammatory state with less Tregs is favoured in lines 124-140.
    • Line 125: Rename the subsection "Tregs in Spontaneous Preterm Birth" to “3. Influence of Tregs on Preterm Birth” to maintain consistency with the new chapter structure.- Thank you for your comment, we have changed the heading title accordingly in line 143.
    • Line 130: Specify the adverse pregnancy conditions and factors mentioned in the text.- We agree with your comment and have since changed it to read the specific adverse pregnancy outcomes: pPROM, chorioamnionitis, bacterial and viral infections, dysbiosis of the cervico-vaginal microbiome, PE and GDM in line 149-151.
    • Line 132: When referring to the "key review," provide the appropriate citation.- We agree and have since specified that the key review references the following (N. Gomez-Lopez, D. StLouis, M. A. Lehr, E. N. Sanchez-Rodriguez, and M. Arenas-Hernandez, “Immune cells in term and preterm labor,” Cell Mol Immunol, vol. 11, no. 6, 2014, doi: 10.1038/cmi.2014.46.) in line 152.
    • To improve readability, organize the results discussed across this chapter in a table to emphasize key findings. Thank you for your comment, rather than creating a table we hope that the improved clarity in the text and the revised graphical abstract allows for the results to be emphasised in a clear manner.
  1. Inflammatory and Immunocompromising Pregnancy Outcomes Associated with sPTB:
    • Rename the current section "Inflammatory and Immunocompromising Pregnancy Outcomes Associated with sPTB" to "3.1. Tregs in Adverse Pregnancy Outcomes Associated with PTB" to align with the chapter reorganization. Thank you for your comment, we have changed the section name to improve the organisation as suggested in line 178-179.  
    • Line 171: Change the level of this section to 3.1.1, titled "Preterm Premature Rupture of Membranes (pPROM)."  We thank you for your comment and have since changed the section heading in line 187.
    • Lines 188-195: This section requires more detail. Additionally, the authors should discuss further research to confirm whether pPROM triggers early T cell maturation during pregnancy.- Thank you for your comment, we have specified how although the study referenced implicates pPROM as a trigger for early T cell maturation, other studies, mentioned in lines 213-215, state this is potentially seen in other forms of sPTB rather than being specific to pPROM.  
    • Line 256: Provide a clear explanation of PBMCs. We agree and have detailed what PBMCs mean by detailing what cell types encompass this sample type in lines 276-277.
  1. Influence of Tregs on Preterm Neonatal Outcomes:
    • Line 380: Add a subtitle to introduce this section more clearly (e.g., 4.1 Sepsis). We agree with your comment and have since added a Sepsis subtitle found in line 403.
    • Lines 392-395: The section would benefit from the inclusion of more neonatal conditions affected by Tregs, such as bronchopulmonary dysplasia and intracerebral hemorrhage, as they can also be influenced by Tregs. Thank you for your comment, we have since included additional sections detailing what has been found in bronchopulmonary dysplasia and intracerebral haemorrhage and how this is relevant to sPTB in lines 479-532.
  1. Discussion:The authors need to ensure consistency throughout the manuscript regarding the objective of the review. Is the focus on the relationship between Tregs and various pregnancy conditions that may influence and lead to sPTB, or, as stated in the title, is the review aimed at exploring the roles of Tregs in spontaneous PTB itself, and how they may contribute to different pregnancy outcomes? This should be clearly articulated in the discussion and aligned with the rest of the manuscript.

Thank you for your comment, we have since added lines 66-68 to ensure that the focus is on tregs in associated conditions which are important in order to help drive research forward. We also aimed to improve clarity surrounding our focus by changing our title to ‘The immunomodulatory role of regulatory T cells in adverse pregnancy outcomes associated with spontaneous preterm birth’.

Round 2

Reviewer 2 Report

Comments and Suggestions for Authors

The authors accepted the proposed revisions and significantly improved the structure of the paper. The main aim of the paper is now clearly formulated: a literature review of the current state of knowledge on the potential roles of regulatory T cells (Tregs) in adverse pregnancy outcomes associated with spontaneous preterm birth.

Title 3.1. needs to be improved. It contains both new and earlier titles that have been mixed together.

Author Response

Title 3.1. needs to be improved. It contains both new and earlier titles that have been mixed together.

Thank you for your comments we have amended accordighly.